# OpenProteinSet: Training data for structural biology at scale

**Gustaf Ahdritz**
Harvard University
gahdritz@g.harvard.edu

**Nazim Bouatta**
Laboratory of Systems Pharmacology, Harvard Medical School
nazim_bouatta@hms.harvard.edu

**Sachin Kadyan**
Columbia University

**Lukas Jarosch**
Columbia University

**Daniel Berenberg**
Prescient Design, Genentech & Department of Computer Science, New York University

**Ian Fisk**
Flatiron Institute

**Andrew M. Watkins**
Prescient Design, Genentech

**Stephen Ra**
Prescient Design, Genentech

**Richard Bonneau**
Prescient Design, Genentech

**Mohammed AlQuraishi**
Department of Systems Biology, Columbia University
m.alquraishi@columbia.edu

## Abstract

Multiple sequence alignments (MSAs) of proteins encode rich biological information and have been workhorses in bioinformatic methods for tasks like protein design and protein structure prediction for decades. Recent breakthroughs like AlphaFold2 that use transformers to attend directly over large quantities of raw MSAs have reaffirmed their importance. Generation of MSAs is highly computationally intensive, however, and no datasets comparable to those used to train AlphaFold2 have been made available to the research community, hindering progress in machine learning for proteins. To remedy this problem, we introduce OpenProteinSet, an open-source corpus of more than 16 million MSAs, associated structural homologs from the Protein Data Bank, and AlphaFold2 protein structure predictions. We have previously demonstrated the utility of OpenProteinSet by successfully retraining AlphaFold2 on it. We expect OpenProteinSet to be broadly useful as training and validation data for 1) diverse tasks focused on protein structure, function, and design and 2) large-scale multimodal machine learning research.

## 1 Introduction

*Multiple sequence alignments* (MSAs) comprise sets of related protein sequences with their amino acid residues in correspondence ("aligned"). MSAs encode rich information about the functional and structural features of a protein family by summarizing the (co-)evolutionary trajectory of its sequence.

MSAs are used in a wide variety of bioinformatic applications, including protein function prediction [3, 4, 5], protein language models [6, 7, 8, 9, 10], disease variant prediction [11, 12], phylogeny [13, 14], protein design [15, 16, 17], protein classification [18], and, most notably, protein structure prediction [19, 25, 26, 27, 28, 29, 30, 31, 32, 20, 21, 22, 23, 24]. Early work on the latter, culminating

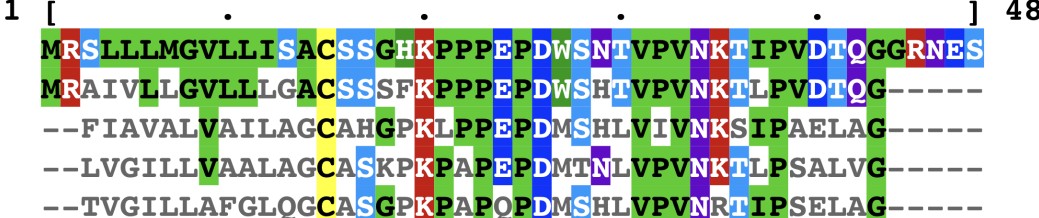

Figure 1: **MSA primer**. Five rows of the OpenProteinSet MSA for PDB protein 3ZBI, chain C [1, 2]. Each row of an MSA is a protein sequence. Proteins are one-dimensional strings composed with a vocabulary of 20 amino acids—or "residues"—each represented by a letter. The target or "query" protein is given in the first row of the MSA. Subsequent rows are evolutionarily related ("homologous") proteins retrieved from a sequence database on the basis of similarity to the query sequence. To improve alignments and accommodate sequences whose length has changed over time, MSA alignment software can insert "gaps" (represented here by dashes) in or delete residues from homologous sequences. Highlights indicate conserved residues. The number of homologous sequences in an MSA ("depth") and their diversity both contribute to the MSA's usefulness.

in the original AlphaFold, achieved notable success by training models on summary statistics derived from MSAs [19, 25, 26, 27, 28, 29, 30, 31, 32, 20, 21]. More recently, large transformer-like neural networks [33] that predict protein structure by directly attending over raw MSAs came to prominence [7]. Among them, AlphaFold2 reached near-experimental accuracy for most proteins at the 14th biannual Critical Assessment of Structure Prediction (CASP) by attending over raw MSAs alongside *structural templates* of homologous proteins [23]. Follow-up work, including RoseTTAFold and the state-of-the-art protein complex structure prediction model AlphaFold-Multimer [24, 34], build on the same techniques. The dependence of these methods on sufficiently deep, diverse MSAs and close structural homologs is evidenced by the fact that they perform worst on proteins that lack them [23].

Despite the central importance of MSAs, the quantity of precomputed MSAs accessible to the research community has not kept pace with the demands of modern machine learning methods. Large models like AlphaFold2 or MSA Transformer [7] were trained on internal datasets of millions of MSAs, and the computation of various official databases of AlphaFold2 predictions [35, 36, 37] would have required hundreds of millions more. None of this data has yet been released to the public, however, and existing public MSA databases [38, 39, 40] are comparatively small and outdated. Raw sequence and structure data are available in large quantities under open licenses [23, 41, 42, 43] and there also exist several mature, open-source software suites for computing MSAs at varying levels of sensitivity [44, 45, 46]. Together, these resources are sufficient to generate MSAs at scale; indeed, they were used to create the aforementioned unreleased datasets. Nevertheless, doing so is computationally expensive. Depending on target sequence length and the size of the sequence database being searched, generating a single MSA with high sensitivity can take several hours. This effectively renders research at the forefront of protein machine learning and bioinformatics inaccessible to all but a few large research groups.

Here, we present OpenProteinSet, a large corpus of precomputed MSAs suitable for training bioinformatic models at the scale of AlphaFold2 and beyond. OpenProteinSet contains an updated reproduction of AlphaFold2's unreleased training set, including MSAs and structural template hits for all unique Protein Data Bank (PDB) chains. It also incorporates more than sixteen million MSAs, computed for each cluster in Uniclust30 [47]. From these, we identify a maximally diverse and deep subset of MSAs that are well-suited for AlphaFold2-style training runs and provide associated AlphaFold2 structure predictions.

We have demonstrated the utility of OpenProteinSet by using it to train OpenFold, a trainable, open-source reproduction of AlphaFold2 [48], achieving accuracy at parity with that of DeepMind's original model. Model parameters resulting from these experiments have been made publicly available.

Not counting these validation experiments or postprocessing, OpenProteinSet represents millions of compute-hours.

After a brief review of related work in Section 2, we provide an overview of the composition of OpenProteinSet in Section 3. Section 4 describes our retraining experiments. We conclude with a discussion in Section 6.

| Sequence origin | Count (approx.) | MSA | Template hits | Structure |
|---|---|---|---|---|
| PDB (all unique chains) | 140,000 | ✓ | ✓ | Experimentally determined |
| Uniclust30 (filtered) | 270,000 | ✓ | ✓ | Predicted by AlphaFold2 |
| Uniclust30 (unfiltered) | 16 million | ✓ | ✕ | ✕ |

Table 1: OpenProteinSet at a glance.

## 2 Related work

**MSAs in structural bioinformatics**: Techniques based on identifying residue-residue correlations in MSAs ("co-variation analysis") are ubiquitous in structural bioinformatics. They have existed in various forms for more than two decades [49, 50], but were initially constrained by the unavailability of sufficient protein sequence data to generate deep MSAs (*i.e.,* comprising many highly diverse sequences). With the onset of next-generation sequencing technology, exponential growth in sequenced genomes and metagenomes led to an explosion in the availability of protein sequence data.

This explosion enabled some of the first successful applications of MSA-based structure prediction methods to proteins [19, 25, 26]. To date, modern machine learning-based approaches rely almost exclusively on MSAs. The first successful models applied residual and convolutional architectures to preprocessed MSA summary statistics [27, 28, 20, 30, 31, 21, 32]. The MSA Transformer was the first to successfully apply transformers to a large corpus (26 million) of unprocessed MSAs in an unsupervised fashion [7], extending prior work on protein language models (PLMs) [10, 8, 51]. Contemporaneously, AlphaFold2 was developed to take MSAs as input to predict protein structures and is additionally trained with an unsupervised BERT-style masked MSA prediction objective [52]. The resulting model, along with its successor AlphaFold-Multimer, has been widely recognized as a revolution in protein structure prediction. Since then, protein structure prediction models that replace MSAs with embeddings from giant PLMs have emerged [22, 9, 53]. They show promise as an emerging technology, but they have so far failed to match the performance of MSA-based methods across the board, significantly underperforming AlphaFold2-based entrants on difficult targets at the most recent installment of CASP [54].

While protein structure prediction is perhaps the most celebrated use case for MSAs, they are broadly used in other areas of bioinformatics. Analogously to natural language processing, unsupervised language modeling of raw MSAs produces rich representations with broad applicability, including in protein design [16], semantic similarity inference [55], and few-shot protein function prediction, where MSA-based models outperformed comparable models trained on individual sequences alone [5]. Long before transformers, summary statistics manually derived from MSAs were already indispensable inputs for diverse tasks ranging from protein classification [18] to disease variant prediction [11, 12].

**MSA software**: There exists a large ecosystem of software for computing MSAs by querying large sequence databases. The commonly used programs HHMer [45] and HHblits [44] are highly sensitive, identifying evolutionarily related proteins with high recall and precision. These tools are slow and memory-intensive, however; they may run for several hours or even days to compute a single MSA. As an alternative, the efficient MMSeqs2 method trades off sensitivity for an order-of-magnitude improvement in runtime and is commonly used for fast inference with AlphaFold2, most notably in ColabFold [56]. Like the MSAs on which AlphaFold2 was trained, OpenProteinSet MSAs are computed with HHMer and HHblits for maximal sensitivity.

**MSA databases**: Responding to the high demand for precomputed MSAs, the community has produced a handful of public MSA repositories. ProteinNet, a repository of standardized protein data for the purposes of machine learning, includes MSAs for approximately 100,000 Protein Data Bank (PDB) protein structures released before May 2016 [40]. Earlier databases are much smaller and less diverse [38, 39]. After the initial release of OpenProteinSet in June 2022, a handful of other open MSA repositories have begun to appear, including PSP, a repository of approximately 1 million MSAs computed with MMSeqs2 [57], and a similar reproduction of about 500,000 MSAs generated according to the procedures outlined in the AlphaFold2 paper [58]. OpenProteinSet is more accurate and larger than any other MSA database.

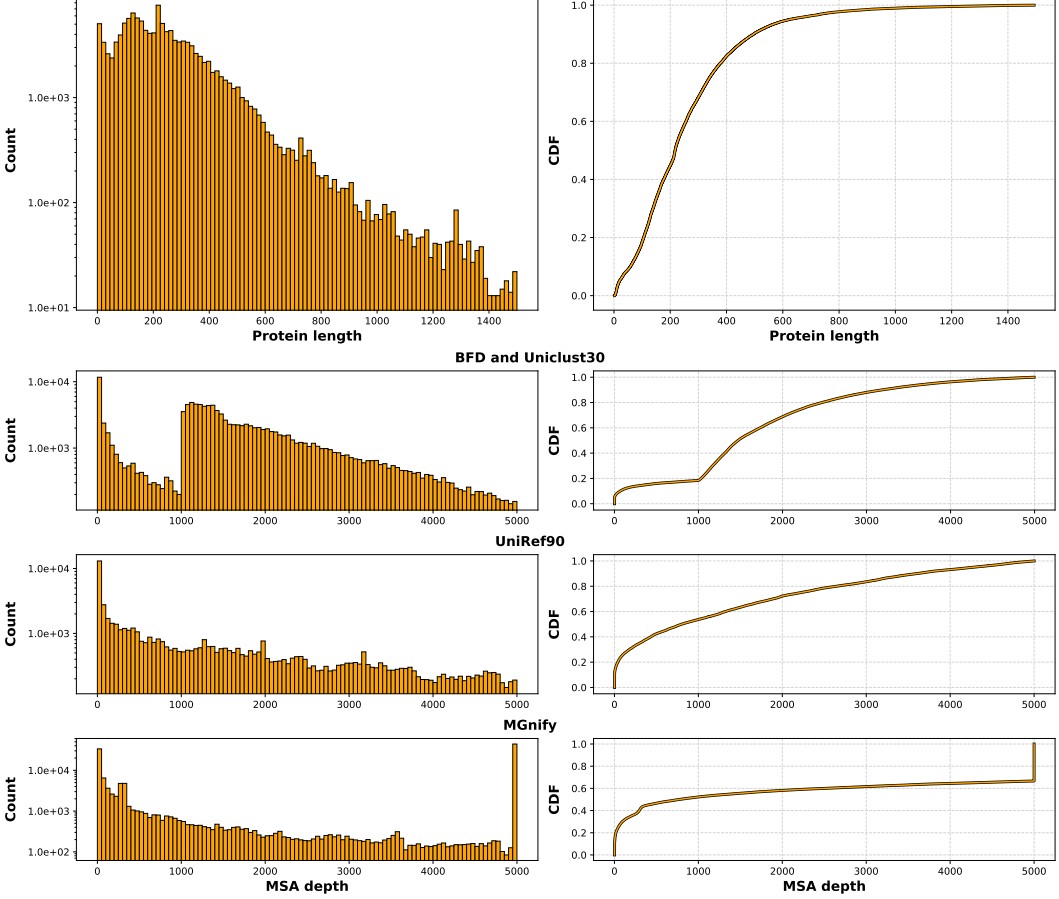

Figure 2: **PDB MSA statistics.** (First row) Number of proteins by sequence length in the PBD portion of OpenProteinSet (left) and the corresponding cumulative density function (CDF) (right). The mean length is 265; the median is 218. (Bottom rows) Depths of MSAs in the PDB portion of OpenProteinSet (left) and the corresponding cumulative density function (CDF) (right). Note that three MSAs are computed for each PDB chain in OpenProteinSet: one using BFD and Uniclust30 (top), one using UniRef90 (middle), and one using MGnify (bottom).

## 3 Methodology

OpenProteinSet consists of more than 16 million unique MSAs generated according to the procedures outlined in the AlphaFold2 paper [23]. This count includes MSAs for all 140,000 unique chains available in the PDB as of April 2022, immediately before the beginning of CASP15, and 16 million MSAs computed for each sequence cluster in Uniclust30 against the same database. From this latter set, we identify 270,000 maximally diverse representative clusters suitable to e.g. serve as the self-distillation set in the AlphaFold2 training procedure. Structural template hits and structure files are also available for this set and all PDB chains.

For each PDB chain, we compute three MSAs using different alignment tools and sequence databases. JackHMMer [45] was used to separately search MGnify [42] and UniRef90 [59]; HHblits-v3 was used to search the Big Fantastic Database (BFD) [23] and Uniclust30 [47]. BFD is a large sequence database prepared for AlphaFold2 that draws its approximately 2.2 billion entries from reference databases, metagenomes, and metatranscriptomes. MgNify (as of 2019) is another environmental database of approximately 300 million sequences. UniRef90 and Uniclust30 are clusterings of UniprotKB [43] proteins at 90% and 30% pairwise sequence identity, respectively, using different clustering algorithms.

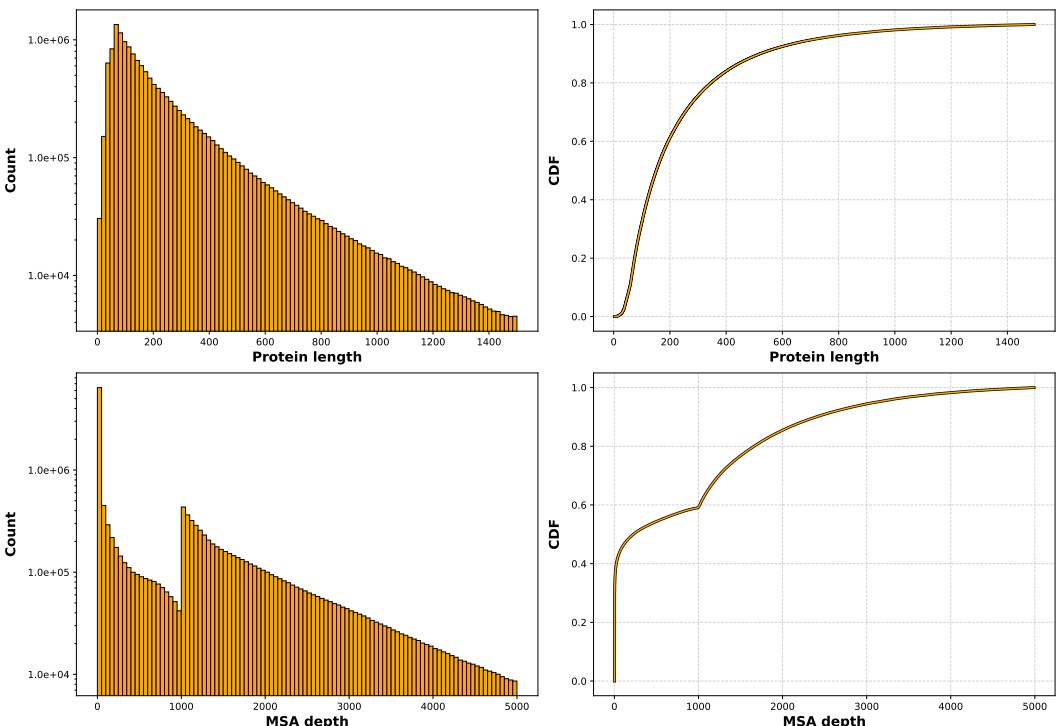

Figure 3: **Uniclust30 MSA statistics.** (Top) Number of proteins by sequence length in the Uniclust30 portion of OpenProteinSet (left) and the corresponding cumulative density function (CDF) (right). The mean length is 255; the median is 153. (Bottom) Depths of MSAs in the Uniclust30 portion of OpenProteinSet (left) and the corresponding cumulative density function (CDF) (right). The average MSA depth is 940; the median is 262.

Structural templates were identified by searching PDB70 [60] using the UniRef90 MSA using HHSearch [44]. Corresponding structures can be retrieved from publicly available PDB mmCIF files using scripts in OpenFold [48].

As in the procedure used to generate AlphaFold2's training set, we changed some of the default options of MSA generation tools. For a list of specific command-line options changed, please consult the supplementary material. One important change is that HHBlits was run for three iterations.

To generate the Uniclust30 MSAs, we performed an all-against-all search on Uniclust30 using HHblits-v3 with the same parameter settings as before. This yielded approximately 16 million MSAs, one for each cluster.

To create a filtered subset of diverse and deep MSAs, we then iteratively removed MSAs whose representative chain appeared in the greatest number of other MSAs. This was repeated until each representative chain appeared only in its own MSA. For parity with the corresponding (unreleased) AlphaFold2 set, we further removed clusters whose representative sequences were longer than 1,024 residues or shorter than 200. Finally, we removed clusters whose corresponding MSAs contained fewer than 200 sequences, leaving just 270,262 MSAs. Template hits were again computed using HHsearch against PDB70. For each representative chain in this subset, we generated structure predictions using OpenFold run with AlphaFold2 weights. Note that, unlike the hundreds of millions of AlphaFold2 predictions made available by DeepMind and EMBL-EBI [35, 36, 37], these are paired with high-quality, diverse MSAs, making it possible to use them as training data for new structure prediction models. All of the above—the 16 million unfiltered Uniclust30 MSAs and filtered-chain template hits and structure predictions—are included in OpenProteinSet.

Overall, the MSAs in OpenProteinSet represent more than four million hours of computation. Its contents are summarized in Table 1.

All MSAs are in A3M format.[1] Template hits are provided in HHSearch's HHR format, while structure predictions are in PDB format. All data is made available under the CC BY 4.0 license.

For all MSAs currently in OpenProteinSet, we used copies of UniRef90 downloaded on December 19, 2021, BFD downloaded on December 20, 2021, Uniclust30 downloaded on December 28, 2021, and MGnify downloaded on January 14, 2022. To compute templates, we used PDB70 downloaded on December 19, 2021. In all cases, we used the most recent versions of each database available at the time. As we update OpenProteinSet with new sequences, we will continually upgrade them.

We used HH-suite version 3.3.0 (commit hash `dc74ac`) and jackhmmer from HMMER3.1.

## 4    Experiments

To demonstrate the utility of OpenProteinSet, we used it as training data for a replication of AlphaFold2, a groundbreaking but previously unreplicated protein structure prediction network trained on raw MSAs. Our AlphaFold2 training code is implemented in OpenFold, our open-source reproduction of the AlphaFold2 training code [48].

First, we simulated the full AlphaFold2 training procedure outlined in Table 4 of the supplement to the AlphaFold2 paper. We used the PDB component of OpenProteinSet as the initial training set and our set of 270,000 filtered Uniclust30 proteins as the self-distillation set. We used a PDB cutoff of December 2021. Training was run on a cluster of 44 A100s. Given the prohibitive costs of training the full model from scratch, original AlphaFold2 weights were used as the pre-distillation model to generate Uniclust30 structure predictions.

To evaluate the resulting OpenFold weights against AlphaFold2, we computed `model_1` predictions for each currently available "all groups" CASP15 domains ($n = 90$) and evaluated them using the GDT-TS score [61]. OpenFold reached a mean score of 73.8 (95% confidence interval = 68.6 - 78.8) while AlphaFold2 reached 74.6 (95% confidence interval = 69.7 - 79.2). Confidence intervals of each mean are estimated from 10,000 bootstrap samples. OpenFold did at least as well as AlphaFold2 on exactly 50% of targets. Superimposed predictions are shown in Figure 4.

Weights from this experiment are available under a permissive license in the OpenFold GitHub repository.[2]

Next, to estimate variance from different weight initializations and other sources of randomness in training, we trained 15 models on the PDB tranche with different seeds for 10,000 initial training steps (compared to more than 75,000 in the full training run), taking advantage of the fact that OpenFold/AlphaFold2 achieves much of its final accuracy relatively quickly (as much as 90% of its final accuracy in less than 3% of the total training time). We observe very little run-to-run variability. For assessment we use lDDT-C$\alpha$ [62], a commonly used accuracy measure for protein structure predictions. We found that on a validation set of 180 unseen CAMEO [63] proteins drawn over a three-month period lDDT-C$\alpha$ was 0.866; the maximum value was 0.881 and the minimum was 0.848, while the median was 0.868. Our final model trained for the full duration on both the PDB and filtered Uniclust30 datasets scores 0.907 on the same validation set.

For more details on both sets of experiments, including specific hyperparameter settings, consult the OpenFold paper [48].

## 5    Limitations

Many centralized sequence databases are rarely updated, and while we used the most recent versions of each wherever possible, most of the MSAs currently in OpenProteinSet were computed in early 2022. Given that the number and diversity of known sequences is continually increasing, this means that OpenProteinSet—like any repository of precomputed MSAs—may age over time and need to be updated for optimal downstream performance. OpenProteinSet entries that currently have shallow MSAs or few structural homologs are particularly "vulnerable" in this regard. While we

---

[1]A3M is a plaintext format consisting of aligned sequences, one per line (as in Figure 1), and comment lines beginning with '>'. Gaps are represented by dashes ('-') and insertions are represented with lowercase residue letters.

[2]URL: https://github.com/aqlaboratory/openfold

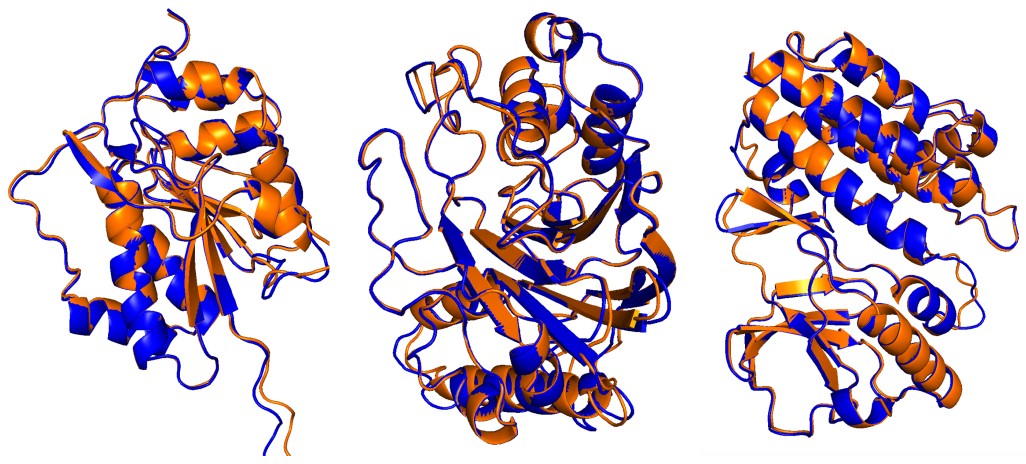

Figure 4: **OpenFold trained with OpenProteinSet reproduces AlphaFold2.** Superimposed Open-Fold (orange) and AlphaFold2 (blue) predictions on three CASP15 domains: from left to right, T1109 (RMSD: 0.306), T1153 (RMSD: 0.263), and T1195 (RMSD: 0.235).

may periodically expand OpenProteinSet with new MSAs, we do not currently plan to update MSAs already in the dataset as new sequences become available.

We note too that we only evaluate OpenProteinSet on monomeric structure prediction and not other popular applications. Nevertheless, the utility of large quantities of MSAs has been firmly established in diverse settings, and we have no reason to believe that OpenProteinSet MSAs in particular will be less useful.

## 6    Discussion

With OpenProteinSet, we have greatly increased the quantity and quality of precomputed MSAs available to the molecular machine learning communities. The dataset has immediate applications to diverse tasks in structural biology. Below, for illustrative purposes, we highlight a handful of additional tasks and settings where we strongly expect high-quality multiple sequence alignments like those in OpenProteinSet to be immediately useful.

**Protein language modeling**: Unsupervised protein language models [8, 64, 22, 7, 65] have become workhorses in the bioinformatic community, as, analogously to natural language models, they encode useful biological knowledge that allows to reason about numerous protein-related tasks. Most are trained on individual protein sequences, but MSA Transformer, a model trained on millions of (unreleased) Uniclust30 MSAs, was able to outperform conventional protein language models on downstream evaluations like protein design, and with fewer parameters [7, 16]. With OpenProteinSet, a dataset of millions of comparable Uniclust30 MSAs, it is now possible for the open-source community to experiment with similar MSA language models, perhaps even in combination with widely available single-sequence data.

**Orphan proteins**: One function of OpenProteinSet is to identify a large number of proteins with few or no known homologs at the time of its creation. "Orphan" proteins like these are often failure cases of models trained on protein data. In protein structure prediction, for example, MSA-based models like AlphaFold2 and RoseTTAFold are known to perform less well on proteins with shallow MSAs [23, 24]. Protein language models are slightly less sensitive to MSA depth in some cases [22], but the gap persists there as well. We expect that a large quantity of additional data on such proteins will be useful to validate and improve bioinformatic methods. Because OpenProteinSet

effectively clusters sequence space, it also enables important validation experiments not possible with unclustered sequences alone, like training on one region of protein space and testing on another.

**Multimodal deep learning**: Beyond bioinformatics, a popular line of deep learning research studies the effects of training extremely large neural networks on data from diverse modalities. While the most commonly studied modality pairing is language and image data [66, 67, 68, 69, 70, 71], unsupervised co-training on additional modalities—including audio [72], robotics tasks [71, 73], and, indeed, raw protein sequence data—has been shown to enrich the knowledge and capabilities of models. Multimodal language models jointly trained on English text and biological sequence data have already been used to identify protein-protein interactions [74], classify adverse reactions to drugs [75], and caption molecules [76]. The multimodal scientific language model Galactica was also trained on protein sequences [77]. More indirectly, protein data often appears as a component in benchmarks for multimodal training methods. It has recently been added to DABS, a multimodal benchmark for unsupervised learning techniques [78, 79], and has been used to study multimodal scaling laws in generative models [80] and test the capabilities of pretrained language models across modalities [81]. As models become increasingly data-hungry, we believe databases like OpenProteinSet will be valuable on both of these fronts, as reservoirs of biological knowledge for generalist multimodal language models and also as tools for the empirical study of multimodal training *per se*.

Overall, we hope that OpenProteinSet will further democratize research in bioinformatics, machine learning on proteins, and beyond.

## Acknowledgments and Disclosure of Funding

We would like to thank the Flatiron Institute for providing computing resources and Amazon Web Services for hosting OpenProteinSet. Individually, we would like to thank Milot Mirdita and Martin Steinegger for their valuable support and expertise.

G.A. is supported by a Simons Investigator Fellowship, NSF grant DMS-2134157, DARPA grant W911NF2010021, and DOE grant DE-SC0022199. N.B. is supported by DARPA PANACEA program grant HR0011-19-2-0022 and NCI grant U54-CA225088. M.A. is a member of the Scientific Advisory Boards of Cyrus Biotechnology, Deep Forest Sciences, Nabla Bio, Oracle Therapeutics, and FL2021-002, a Foresite Labs company.

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
