# A Supplementary materials

## A.1 Access instructions

OpenProteinSet is hosted by the Registry of Open Data on AWS (RODA) and can be accessed at the following link: registry.opendata.aws/openfold/.

## A.2 Documentation and intended uses

We include a datasheet [1] in Section B. Detailed documentation on the precise structure and content of the dataset is provided on the dataset's landing page.

## A.3 Data format

All OpenProteinSet files are in standard plaintext formats (A3M for MSAs, HHSearch format for template hits, and PDB for structure files) that can be read by a wide variety of bioinformatics software.

## A.4 Code

Code for the training experiments referenced in this paper, including scripts for downloading and parsing OpenProteinSet, can be found in the OpenFold GitHub repository (github.com/aqlaboratory/openfold).

## A.5 License

OpenProteinSet is made available under the CC BY 4.0 license. A copy of the license is provided with the dataset. The authors bear all responsibility in case of violation of rights.

## A.6 Hosting plan

OpenProteinSet will continue to be hosted on RODA for the foreseeable future. Should that change, we will create an issue on the OpenFold GitHub repository.

## A.7 Alignment tool settings

For JackHMMer, we used

```
-N 1 -E 0.0001 -incE 0.0001 -F1 0.0005 -F2 0.00005 -F3 0.0000005
```

and then capped outputs at depth 5000. For HHBlits, we used

```
-n 3 -e 0.001 -realign_max 100000 -min_prefilter_hits 1000
              -maxfilt 100000 -maxseq 1000000
```

# B Datasheet

## B.1 Motivation

**For what purpose was the dataset created?**

The dataset was created to meet growing demand for precomputed protein alignment data. This data can be used for a wide variety of protein-related tasks in structural bioinformatics, including protein structure prediction, protein design, protein language modeling, and more.

**Who created the dataset (e.g., which team, research group) and on behalf of which entity (e.g., company, institution, organization)?**

OpenProteinSet was created by the AlQuraishi Lab at Columbia University.

**Who funded the creation of the dataset?**

The computational resources used to produce the alignments in OpenProteinSet were generously provided by the Flatiron Institute.

Author Nazim Bouatta is supported by DARPA PANACEA program grant HR0011-19-2-0022 and NCI grant U54-CA225088.

**Any other comments?**

No

### B.2 Composition

**What do the instances that comprise the dataset represent (e.g., documents, photos, people, countries)?**

The instances of OpenProteinSet are sets of alignment data corresponding to protein sequences. These include multiple sequence alignments (MSAs) in A3M format, template hits in HHSearch format, and structure files in PDB format.

**How many instances are there in total (of each type, if appropriate)?**

OpenProteinSet contains three MSAs for each of the approx. 140,000 proteins in the Protein Data Bank (PDB) as of May 2022 and one MSA for each of about 16 million Uniclust30 clusters. All PDB proteins and approximately 270,000 Uniclust30 clusters come with template hits. The latter 270,000 also include AlphaFold2 structure predictions (structures for the PDB proteins are available directly from PDB).

**Does the dataset contain all possible instances or is it a sample (not necessarily random) of instances from a larger set?**

We include MSAs for all PDB chains and all Uniclust30 clusters. All PDB chains come with template hits, and experimentally determined structures for each can be found on PDB. For computational reasons, we only provide template hits and structure predictions for 270,000 maximally diverse Uniclust30 clusters (out of the full 16 million). Note that the UniProtKB proteins that make up Uniclust30 clusters lack experimentally determined structures.

**What data does each instance consist of?**

OpenProteinSet instances consist of raw MSAs in A3M format, template hits in HHSearch format, and/or structure predictions in PDB format.

**Is there a label or target associated with each instance?**

No.

**Is any information missing from individual instances?**

No.

**Are relationships between individual instances made explicit (e.g., users' movie ratings, social network links)?**

N/A

**Are there recommended data splits (e.g., training, development/validation, testing)?**

No.

**Are there any errors, sources of noise, or redundancies in the dataset?**

The authors are not aware of any errors in the dataset. While the set contains no MSAs for duplicate sequences, there inevitably exist pairs of evolutionarily related MSAs in the dataset with high degrees of overlap, which may be redundant in certain contexts. Note that we provide a split of 270,000 Uniclust30 MSAs filtered for maximal diversity for this reason.

**Is the dataset self-contained, or does it link to or otherwise rely on external resources (e.g., websites, tweets, other datasets)?**

The dataset is mostly self-contained, but structure data for the PDB portion of the dataset is not included. Experimentally determined structures for each of these proteins will be publicly accessible for the foreseeable future under the corresponding entry of PDB (https://www.rcsb.org/).

**Does the dataset contain data that might be considered confidential (e.g., data that is protected by legal privilege or by doctor-patient confidentiality, data that includes the content of individuals' non-public communications)?**

No. OpenProteinSet is built using publicly available databases of protein sequences, of which the overwhelming majority are nonhuman.

**Does the dataset contain data that, if viewed directly, might be offensive, insulting, threatening, or might otherwise cause anxiety?**

No.

**Does the dataset relate to people?**

No.

## B.3 Collection process

**How was the data associated with each instance acquired?**

With the exception of AlphaFold2 structure predictions, all data in OpenProteinSet is ultimately derived from raw data from publicly accessible protein databases. Target sequences were drawn from PDB and UniProtKB via Uniclust30. MSAs were constructed by searching over the Big Fantastic Database (BFD), UniRef90, and MGnify. Template hits are computed with PDB70. For precise data generation procedures, please consult the OpenProteinSet paper.

**What mechanisms or procedures were used to collect the data (e.g., hardware apparatus or sensor, manual human curation, software program, software API)?**

We chose sequence databases according to the procedure outlined in the AlphaFold2 paper [2]. We did not collect novel protein data ourselves.

**If the dataset is a sample from a larger set, what was the sampling strategy (e.g., deterministic, probabilistic with specific sampling probabilities)?**

N/A

**Who was involved in the data collection process (e.g., students, crowdworkers, contractors) and how were they compensated (e.g., how much were crowdworkers paid)?**

We did not employ external crowdworkers or contractors to construct OpenProteinSet.

**Over what timeframe was the data collected?**

With the exception of data for a handful of new PDB entries, most of the data in OpenProteinSet was constructed using versions of aforementioned sequence databases downloaded in December 2021. As we add new data over time, we expect to upgrade sequence databases to their most recent versions.

**Were any ethical review processes conducted (e.g., by an institutional review board)?**

No.

**Does the dataset relate to people?**

No.

## B.4 Preprocessing/cleaning/labeling

**Was any preprocessing/cleaning/labeling of the data done (e.g., discretization or bucketing, tokenization, part-of-speech tagging, SIFT feature extraction, removal of instances, processing of missing values)?**

No. OpenProteinSet provides raw MSAs, structural template hits, and structure predictions for all unique and contemporaneous PDB sequences and Uniclust30 clusters.

### B.5 Uses

**Has the dataset been used for any tasks already?**

Yes. We have used the dataset to successfully train OpenFold, an open-source reproduction of the state-of-the-art protein structure predictor AlphaFold2. The weights from this experiment have been released publicly and are hosted in the OpenFold GitHub repository.

**Is there a repository that links to any or all papers or systems that use the dataset?**

Not at the present time.

**What (other) tasks could the dataset be used for?**

While we constructed it for protein structure prediction, MSAs are sufficiently important primitives in structural bioinformatics that we expect OpenProteinSet will be useful for practically any protein-related machine learning task, including but not limited to protein design, protein language modeling, and protein function prediction.

**Is there anything about the composition of the dataset or the way it was collected and preprocessed/cleaned/labeled that might impact future uses?**

Given that the number of known protein sequences is growing at a rapid pace, OpenProteinSet MSAs will eventually become outdated, at least for certain applications. It is possible that we'll periodically recompute and expand the database, but users should be cognizant of the fact that e.g. proteins that appear as orphans in OpenProteinSet may be closely related to sequences in more recent versions of sequence databases.

**Are there tasks for which the dataset should not be used?**

No.

### B.6 Distribution

**Will the dataset be distributed to third parties outside of the entity (e.g., company, institution, organization) on behalf of which the dataset was created?**

The full dataset is already publicly accessible on the Registry of Open Data on AWS (RODA) (https://registry.opendata.aws/openfold/).

**How will the dataset will be distributed (e.g., tarball on website, API, GitHub)?**

The full dataset is already publicly accessible on the Registry of Open Data on AWS (RODA) (https://registry.opendata.aws/openfold/).

**When will the dataset be distributed?**

The dataset is already available.

**Will the dataset be distributed under a copyright or other intellectual property (IP) license, and/or under applicable terms of use (ToU)?**

OpenProteinSet uses the CC BY 4.0 license.

**Have any third parties imposed IP-based or other restrictions on the data associated with the instances?**

No.

**Do any export controls or other regulatory restrictions apply to the dataset or to individual instances?**

No.

### B.7 Maintenance

**Who is supporting/hosting/maintaining the dataset?**

The dataset is hosted on RODA by AWS and maintained by the AlQuraishi Lab.

**How can the owner/curator/manager of the dataset be contacted (e.g., email address)?**

Recent contact information can be found on the dataset's landing page on RODA. Alternatively, issues can be raised on the OpenFold GitHub page.

**Is there an erratum?**

Not at the present time. Future errata will be published on the dataset's landing page on RODA.

**Will the dataset be updated (e.g., to correct labeling errors, add new instances, delete instances)?**

We may sporadically update the dataset with MSAs for new PDB sequences, but we do not currently have any plans to update MSAs already in the database. If that changes, we'll post updates on the OpenFold GitHub page.

**If the dataset relates to people, are there applicable limits on the retention of the data associated with the instances (e.g., were individuals in question told that their data would be retained for a fixed period of time and then deleted)?**

N/A.

**Will older versions of the dataset continue to be supported/hosted/maintained?**

We expect any future changes to be additive. If that changes, we will communicate our versioning policy on the dataset's landing page on RODA.

**If others want to extend/augment/build on/contribute to the dataset, is there a mechanism for them to do so?**

We accept feedback in the issues of the OpenFold GitHub page.