# OpenReview forum: "OpenProteinSet: Training data for structural biology at scale"
_NeurIPS.cc/2023/Track/Datasets_and_Benchmarks — NeurIPS 2023 Datasets and Benchmarks Poster_

### Official Review · Reviewer_yCQz · 2023-07-02
**A useful data collection for protein sequences and structures**

**Rating:** 6
**Confidence:** 4
**Clarity:** The paper is fairly well written.

**Strengths:**

1. The collection of multiple sequence alignments that authors have curated appears to be soundly constructed, and their computational experiments in retraining OpenFold seem to confirm this point.
2. Seemless integration of these alignments with OpenFold is a useful feature for future users of this collection of alignments.
3. The authors' analysis of their collected protein sequence data, with further discussion in the manuscript, may be useful to users of this alignment dataset.

**Additional Feedback:**

This reviewer would like to emphasize that the collection of MSAs the authors have curated will likely be useful for the research community. However, this reviewer does not believe the manuscript in its current form will be particularly useful for machine learning researchers and practitioners.

**Correctness:**

The multiple sequence alignment dataset the authors propose in this manuscript appears to be constructed in a sound way, by way of industry-standard tools and data repositories for generating alignments.

**Documentation:**

The authors provide sufficient detail describing how they have curated and organized their collection of multiple sequence alignments.

**Ethics:**

There are no ethical concerns that warrant further review.

**Limitations:**

The authors discuss how multiple sequence alignment datasets may soon become out-of-date due to the rapidly-growing nature of protein sequence repositories. They also discuss how they only evaluate the effectiveness of their proposed multiple sequence alignment dataset through monomeric protein structure prediction. However, it remains to be shown how chain-paired multiple sequence alignments can specifically benefit multimeric protein structure prediction and related tasks. As such, it is currently unclear whether the authors' proposed multiple sequence alignments will prove useful outside of the task of monomeric structure prediction.

**Opportunities For Improvement:**

1. Introducing benchmarks **outside** of monomeric structure prediction would go a long way toward encouraging others to invest effort in adopting large collections of multiple sequence alignments such as the one proposed by the authors. For example, it is currently unclear whether using multiple sequence alignments would be useful for tasks outside of structure prediction such as protein design, as the cost of adopting multiple sequence alignments within one's proposed methodology for such a task can often outweigh the benefits of such kinds of data (e.g., large storage and compute requirements w/ computational throughput degradation).

2. The manuscript currently reads as a manual describing how the authors collected protein sequence data, aligned it, and trained OpenFold for monomeric structure prediction using such alignments. In many regards, the results are less than surprising: large quantities of multiple sequence alignments are important for achieving strong performance for structure prediction (as previously shown with AlphaFold). Unfortunately, in its current state, the manuscript does not offer readers many new insights into what they can realistically do with such multiple sequence alignments besides (nearly) reproducing the results of AlphaFold 2 using the OpenFold GitHub repository. As such, without the inclusion of additional computational benchmarks for such alignments (e.g., protein design or multimeric structure prediction), the manuscript currently does not offer readers much value besides pointing them to OpenFold.

3. Most importantly, it seems as though an important point has been missed in the current version of the manuscript: What is the primary insight (or insights) offered to readers as a result of this work? It currently appears to be that "having collections of multiple sequence alignments is important for protein structure prediction", however, this point has previously been made by AlphaFold and similar prior efforts. Said another way, this reviewer believes it would be useful to authors to try to answer the question, "What kind of new science or scientific tasks does this collection of multiple sequence alignments definitively enable?".

4. Lastly, it would be helpful if the authors could expand the discussion of their data analysis results for the sequences and structures they collected accordingly. Currently, it seems like the authors simply plot data analysis figures without identifying any interesting takeaways for readers.

**Relation To Prior Work:**

The authors sufficiently describe the placement of their work in relation to prior works in the literature.

**Summary And Contributions:**

The authors introduce a large collection of multiple sequence alignments for monomeric protein structure prediction. They demonstrate that retraining OpenFold using such multiple sequence alignments allows OpenFold to (nearly) match the performance of AlphaFold 2 in certain benchmark experiments. However, other computational experiments with such multiple sequence alignments have not been performed, which currently limits the impact that this work can have outside of monomeric structure prediction.

---

> ### Author Response · Authors · 2023-08-10
>
> We thank the reviewer for their review! We are glad they believe our work “will likely be useful for the research community” and that its integration with OpenFold will be helpful.
>
> We have updated our manuscript in line with some of the concerns expressed by the reviewer. Below, we address their questions and concerns individually.
>
> **Re: Benchmarks outside monomeric structure prediction**
>
> >As we argue in our response to reviewer GyWH, we decided not to include additional benchmarks for the following two reasons:
>
> >1. The utility of MSAs is well-established in bioinformatics, and they have been in continuous use across bioinformatic applications like protein folding, protein function prediction, disease variant prediction, protein classification and phylogeny, and protein design for over a decade. As in other areas of ML, the value of large-scale unsupervised learning on MSAs has also been demonstrated fairly conclusively (AlphaFold2, AlphaFold-Multimer, MSA Transformer (Rao et al. 2021), etc.). Since we use standard tools and datasets to create our MSAs, and since we were able to use our MSAs to replicate one of the most important modern MSA-based models (AlphaFold2), we have no reason to expect that OpenProteinSet MSAs will be less effective in other settings.
> >2. As we argue in the manuscript, the generation of MSAs and the AlphaFold2 experiments were immensely computationally expensive (requiring millions of CPU hours and ~100k A100 hours, respectively). In general, unsupervised learning on MSAs is intensive, since MSAs, like images, are two-dimensional. OpenProteinSet is also very large, and any additional experiment demonstrating its utility would be expensive for that reason as well.
>
> >In protein design, MSAs are already very prevalent (see “Data-driven computational protein design” (Frappier & Keating 2021)). Large-scale unsupervised learning on MSAs for protein design is also argued to be effective in (Sgarbossa et al. 2023), cited in the most recent revision of our manuscript. While it is true that multiple sequence alignments may be more computationally expensive than other approaches, this does not prevent their use in practice, and we feel that quantifying the tradeoffs involved is 1) difficult to do convincingly at a high level and also 2) beyond the scope of a datasets and benchmarks paper.
>
> >On the question of multimeric protein structure prediction, we feel strongly that there is already good reason to believe that OpenProteinSet will be useful in this setting. AlphaFold-Multimer, DeepMind’s follow-up to AlphaFold2, successfully applies a lightly postprocessed version of the AlphaFold2 dataset and an almost identical architecture to the multimeric setting and remains the state of the art today. The similarity of our dataset construction methodology to DeepMind’s and our strong AlphaFold2 reproduction results are good evidence that OpenProteinSet could be used in the same way. While we would ideally include an AlphaFold-Multimer reproduction as well, this would be extremely time-consuming and expensive (the AlphaFold2 reproduction consumed more than a year of full-time development and a huge amount of A100 time), and we do not believe that it would contribute enough to the manuscript to warrant the cost.
>
> **Re: Lack of insights in the manuscript**
>
> >We agree with the reviewer that the original manuscript placed too much emphasis on monomeric protein structure prediction. While that application is an important one, MSAs are known to be useful in almost every task that touches on proteins. We have updated the manuscript to reflect this fact. We have expanded the related works section with a more extensive discussion of other tasks. We have also expanded the discussion section to include three applications where we strongly expect OpenProteinSet to be immediately useful: large-scale protein language modeling, studies of orphan proteins, and multimodal deep learning (we expect this to be interesting to non-bioinformatic researchers as well). This does not include obvious applications in other aforementioned tasks. Finally, as suggested by another reviewer, we have made the manuscript more friendly to readers not already familiar with bioinformatics. We hope that this provides a springboard for future research and improves the value of the manuscript itself.
>
> **Re: Expanded data analysis**
>
> >We have improved the presentation of the data statistic figures (which were previously separated into multiple figures and difficult to interpret) and added more digestible summary statistics in the captions of each. We hope these provide a better record of the contents of OpenProteinSet. If there are other analyses the reviewer would like to see included, we would be happy to oblige.
>
>
> Please let us know if there remain any unaddressed concerns standing in the way of a more positive endorsement of OpenProteinSet. We look forward to working with the reviewer to further improve the quality of the manuscript.

---

> > ### Comment · Reviewer_yCQz · 2023-08-12
> > **Response to Authors' Latest Revisions**
> >
> > I want to thank the authors for their latest revisions to the manuscript. I feel as though it has been notably improved since its initial submission, especially regarding its related works section and citations, its MSA primer for the non-bioinformatic readers, and the refactoring of its data figures. Also important is its expanded discussion on reasonably-tractable use cases for OpenProteinSet beyond monomeric protein structure prediction. Accordingly, I would like to raise my review score to a 6 ("Marginally above acceptance threshold").

---

> > > ### Author Response · Authors · 2023-08-15
> > >
> > > Thanks! We're glad we were able to address some of your concerns.

---

> > > > ### Author Response · Authors · 2023-08-16
> > > >
> > > > Could we trouble the reviewer to update their score in the original review as well? Not sure if this is strictly necessary, but you never know :)

---

> > > > > ### Comment · Reviewer_yCQz · 2023-08-16
> > > > > **Response to Score Update Request**
> > > > >
> > > > > Score updated.

---

### Official Review · Reviewer_S8Br · 2023-07-20
**Good dataset, unsure about novelty**

**Rating:** 6
**Confidence:** 4
**Correctness:** Dataset curation is sound. Evaluation…
**Clarity:** Paper is written very clearly.

**Strengths:**

- Very large scale
- Convincing use case

**Additional Feedback:**

-

**Documentation:**

- Maintenance: is hosting on AWS guaranteed for a longer time span, or is this subject to a grant/bill covered by the authors?

**Ethics:**

No ethical concerns.

**Limitations:**

The authors point out that MSAs can become outdated in the sense that with growing sequence DBs MSAs can become deeper or individual sequences might change. Is there any plan for OpenProteinSet to be a resource that gets updates, or is it a one-off static dataset? Both are valid, just would be good to know.

**Opportunities For Improvement:**

- Long/short sequence filtering: From the presented text, I don’t understand how these steps serve the aim of getting a “subset of diverse and deep MSAs”
- Are there potential use cases beyond structure prediction and LM training?

I am worried whether this constitutes an attempt at double publishing or not. As I understand it, OpenProteinSet is the training set of the OpenFold method, and as such is also described in the OpenFold preprint. Normally of course I would not take preprints into account but given that the authors themselves cite the OpenFold preprint here, I have to assume it’s considered a separate work and that they intend to publish that preprint separately, rather than this manuscript superseding the preprint. The OpenFold preprint that is cited explicitly introduces OpenProteinSet as a contribution. I cite from the abstract:
Here we report OpenFold, a fast, memory-efficient, and trainable implementation of AlphaFold2, and OpenProtein-Set, the largest public database of protein multiple sequence alignments.

Admittedly I have not encountered this situation before (a manuscript citing a preprint that describes the contribution of the manuscript) and am unsure how to evaluate this. Just to be absolutely clear, my confusion stems from the fact that the preprint is cited here, not the fact that it exists.  I would like to ask the authors to clarify, and ultimately ask the AC to give their opinion.


**Relation To Prior Work:**

Other available MSA databases are discussed.

**Summary And Contributions:**

The paper describes OpenProteinSet, a massive set of multiple sequence alignments (MSAs)  of proteins. Given that SOTA structure prediction algorithms rely on MSAs, and computing MSAs is expensive, this is undoubtedly a valuable resource.
The main contribution of OpenProteinSet can be understood as its massive scale and open availability, removing the need for researchers to attempt to generate such a large MSA set themselves. The methodology is adequate and based on what is laid out in AlphaFold.

---

> ### Author Response · Authors · 2023-08-10
>
> We thank the reviewer for their review! We are pleased they think OpenProteinSet is a “valuable resource” and that the manuscript is written clearly. We address concerns/questions line-by-line below.
>
> **Re: Concerns of double publishing**
>
> >The reviewer rightly points out that the OpenProteinSet manuscript cites the OpenFold preprint, which in turn presents an early version of OpenProteinSet as a contribution. OpenFold is currently under review elsewhere, and as part of that process we recently decided to separate the OpenFold codebase and reproduction/ablation experiments from OpenProteinSet, since the dataset represents a huge computational effort in its own right and also has applications far beyond protein folding. While we would like to note that the OpenProteinSet manuscript under review here describes a newer, much larger version of OpenProteinSet than the one presented in the most recent OpenFold preprint from November 2022, we agree with the reviewer that there is apparent overlap between the two manuscripts. We are taking the following steps to make the separation of OpenFold and OpenProteinSet as clear as possible:
>
> >1. We have already submitted the most recent revision of the OpenProteinSet manuscript to arXiv.
> >2. As soon as arXiv posts OpenProteinSet, we will update the OpenFold preprint and its abstract to remove any description of OpenProteinSet and cite the new OpenProteinSet preprint instead. This change will be reflected both in the public OpenFold preprint and in any future published version of the OpenFold paper.
>
> >We take double publishing very seriously and hope that these changes are to the satisfaction of the reviewer.
>
> **Re: Update frequency**
>
> >We have added details on update frequency in the limitations section. These details are 	also provided in the accompanying datasheet. While we plan to add MSAs for new PDB sequences as they are uploaded, we do not currently have any plans to update MSAs already in the dataset, but expect to do so on occasion in the future
>
> **Re: Maintenance**
>
> >OpenProteinSet is generously hosted by AWS at no cost as part of the RODA program. It is not subject to a grant/bill covered by the authors.
>
> **Re: Length filtering**
>
> >The reviewer is right that the length filtering (something we do for parity with the AF2 training set) does not contribute to the aim of producing a “subset of diverse and deep MSAs.” We have reworded that section of the manuscript to reflect that.
>
> **Re: Potential use cases**
>
> >We have expanded the discussion of applications of OpenProteinSet beyond protein structure prediction and bioinformatics. We include protein design, protein function prediction, disease variant prediction, protein phylogeny, studies of orphan proteins, and protein classification. We also discuss the potential of OpenProteinSet as a resource for non-bioinformatic multimodal deep learning research. MSA data is complex and information-rich but very different from traditional image/text/audio data. Biological 	sequence data is already being incorporated into generalist multimodal papers both as a source of biological knowledge (see Meta’s scientific language model Galactica) and also as a useful benchmark for multimodal training techniques (see DABS 2.0).
>
> We hope that we have addressed all of the reviewer’s concerns and questions (if not, please do not hesitate to let us know).

---

> > ### Author Response · Authors · 2023-08-12
> >
> > Update re. double publishing: the OpenProteinSet preprint is now on arXiv (https://arxiv.org/abs/2308.05326) and the OpenFold preprint has been updated (https://www.biorxiv.org/content/10.1101/2022.11.20.517210). It no longer claims OpenProteinSet as a contribution and cites the OpenProteinSet preprint instead.

---

> > > ### Comment · Reviewer_S8Br · 2023-08-16
> > >
> > > Thank you for addressing my concerns! My double publishing concerns are hereby void. Score is updated.

---

### Official Review · Reviewer_GyWH · 2023-07-20
**Review of OpenProteinSet**

**Rating:** 6
**Confidence:** 3
**Clarity:** Yes

**Strengths:**

- The paper introduces a large set of protein MSAs, which could have a considerable impact on the community.
- The generation of MSAs is computationally costly and the authors used a sophisticated protocol similar to the AlphaFold2 procedure.
- The structure information linked to each MSA is a very valuable addition.
- Generally, the paper describes high-quality research.


**Additional Feedback:**

Yes

**Correctness:**

Yes. The dataset construction appears sound.


**Documentation:**

Yes

**Limitations:**

Yes

**Opportunities For Improvement:**

Besides the description of the dataset, the paper does not offer lots of insights. This is probably mainly due to the OpenFold paper being already published elsewhere, but the reviewer would have wished to see some new experiments.


**Relation To Prior Work:**

Yes

**Summary And Contributions:**

The paper introduces OpenProteinSet, a large-scale dataset of protein MSAs with structure information. The set is the largest protein set to date and was already successfully used to train OpenFold to achieve a similar performance as AlphaFold2. Generally, the dataset is very valuable for the community, in particular, due to the provided MSAs which require immense computing.

---

> ### Author Response · Authors · 2023-08-10
>
> We would like to thank the reviewer for their time and are happy they found our work to be “high-quality research.” We discuss their concerns below:
>
> **Re: New insights**
>
> >The reviewer commented that, while the dataset itself is “very valuable” and likely to “have a considerable impact on the community,” they would have liked to see additional 	insights and new experiments in the manuscript. We did not include additional 	experiments for two reasons:
>
> >1. The utility of MSAs is well-established in bioinformatics, and they have been in continuous use across bioinformatic applications like protein folding, protein function prediction, disease variant prediction, protein classification and phylogeny, and protein design over a decade. As in other areas of ML, the value of large-scale unsupervised learning on MSAs has also been demonstrated fairly conclusively (AlphaFold2, MSA Transformer, etc.). Since we use standard tools and datasets to create our MSAs, and since we were able to use our MSAs to replicate one of the most important modern MSA-based models (AlphaFold2), we have no reason to expect that OpenProteinSet MSAs will be less effective in other settings.
> >2. As we argue in the manuscript, the generation of the MSAs and the AlphaFold2 experiments were immensely computationally expensive (requiring millions of CPU hours and ~100k A100 hours, respectively). In general, unsupervised learning on MSAs is intensive, since MSAs, like images, are two-dimensional. OpenProteinSet is also very large, and any convincing experiment proving its utility would be expensive for that reason as well.
>
> >We have, however, updated the manuscript to emphasize additional settings where OpenProteinSet might be useful. We expanded the related works section, decreased the manuscript’s previous emphasis on protein structure prediction, and in the discussion outline applications beyond bioinformatics where we strongly expect OpenProteinSet to be useful. These provide new insights for readers and identify clear avenues for future work. We hope our changes improved the manuscript to the satisfaction of the reviewer.
>
> We recognize that these arguments and improvements are somewhat subjective, and we would be happy to discuss further changes.

---

> > ### Comment · Reviewer_GyWH · 2023-08-29
> > **Comment by Reviewer**
> >
> > Thanks for your answer!

---

### Official Review · Reviewer_UMX4 · 2023-07-25
**Large open dataset of aligned proteins**

**Rating:** 7
**Confidence:** 4
**Correctness:** No issues with correctness.

**Strengths:**

* It is quite apparent that this will be a very useful resource for the protein folding community and the bioinformatics community at large.
* Obtaining this resource at such a scale is computationally inaccessible to many researchers, adding to its value.
* The paper is well-written and technically sound.

**Additional Feedback:**

-

**Clarity:**

* Good clarity overall, except for the suggestion to make some of the bioinformatics terms easier for an ML audience.
* The meaning of command line arguments of JackHMMer in line 121 and 123 are not generally known. These should be explained or moved to supplemental.

**Documentation:**

Adequate documentation and examples are provided.

**Ethics:**

No ethics concerns

**Limitations:**

The match to the venue is not completely clear to me in the sense that this may only be relevant to those in the protein folding community and not ML at large. The authors do not discuss how the ML community can make use of this data apart from training protein structure prediction models. In its current form the paper reads more like it would be most relevant to the bioinformatics community. Perhaps the authors can discuss in more depth how the MSAs can be used in other ML tasks and how to incorporate them into learning models.

**Opportunities For Improvement:**

Some language will be unfamiliar to ML community and should be introduced a bit more extensively (‘homology’, ‘MSA depth’). Perhaps include an illustrative figure of an MSA or define it more formally.

**Relation To Prior Work:**

* Previous work is thoroughly reviewed.

**Summary And Contributions:**

This is a very large open dataset (the largest to date) of multiple-sequence alignments (MSA) of protein sequences. Among many other biological applications, MSAs are an important part of training AlphaFold-style protein structure prediction models however until this point there were no large-scale MSA datasets available at the scale needed to train models that can compete with AlphaFold. The authors use established alignment tools to build these alignments which are to be made publicly available. Access to large datasets of MSAs will allow the community to train their own structure prediction models and potentially generate improvements in structure prediction beyond current SOTA, as well as study the behaviour of these models in an open manner. As a demonstration, the authors train an OpenFold model with their MSAs to achieve comparable performance to the original AlphaFold model trained on a closed set of MSAs.

Overall, this is a strong contribution. My only major concern is the fit to the venue. Apart from those in the ML community working on AF-style folding and bioinformaticians it is not directly clear how the broader ML community can easily make use of this data.

---

> ### Author Response · Authors · 2023-08-10
>
> We are grateful for the reviewer’s constructive comments and are glad they found the work valuable! We attempt to address their individual comments/concerns below:
>
> **Re: Clarity**
>
> >We have made substantial edits to the manuscript to accommodate readers who might not be as familiar with bioinformatics. We took the reviewer’s suggestion to include an illustrative figure of an MSA (Figure 1 in the newest revision). It includes a sample MSA, a brief description of what MSAs and proteins are, and definitions for key terms like “homologous” and “depth.” We moved lists of obscure command-line arguments to the appendix. We also elaborated on lists of related tasks in bioinformatics in the “related works” section.
>
> **Re: Venue fit**
>
> >While we acknowledge that our paper is certainly most valuable to practitioners of machine learning in bioinformatics, we believe the most recent revision of our OpenProteinSet manuscript is a good fit for NeurIPS for three reasons:
>
> >1. The bioinformatic machine learning community is large and growing, especially in the wake of AlphaFold2. For instance, at NeurIPS 2022, there were five papers with “protein” in the title, another three with “molecule,” and one more with “biological sequences” (this is likely a severe undercount of all biology-related papers). Previous protein-related NeurIPS papers have been very successful with hundreds of citations (e.g. “Language models enable zero-shot prediction of the effects of mutations on protein function” in 2021, “Evaluating Protein Transfer Learning with TAPE” in 2019, and "Generative Models for Graph-Based Protein Design" also in 2019). Furthermore, the official NeurIPS call for papers includes a category for "Machine learning for sciences (e.g. climate, health, life sciences, physics, social sciences)."
> >2. We have expanded the discussion in the paper to explain why datasets like OpenProteinSet are of interest to generalist machine learning researchers. Biological sequence data is increasingly used in multimodal machine learning research, both as a way to imbue language models with biological knowledge and also as a way to test and improve multimodal training techniques. Meta’s scientific language model Galactica (Taylor et al. 2022) was trained on protein sequences, among others (see the discussion section in the paper). On the multimodal benchmark side, DABS 2.0 (Tamkin et al. 2022), published in last year’s NeurIPS Datasets and Benchmarks track, includes protein sequence data. As multimodal models grow and become more sophisticated, we believe the complex, unique data in large-scale resources like OpenProteinSet will grow more important as complements to traditional text, audio, and image datasets.
> >3. We have edited the manuscript to decrease its undue emphasis on protein folding and emphasize other use cases in bioinformatics, like protein language modeling, disease variant prediction, protein design, and studies on low-resource orphan proteins. While protein folding is the task that motivated our lab to create the dataset, MSAs are in fact used throughout bioinformatics, giving OpenProteinSet a broad appeal.
>
> Please do let us know if we have adequately addressed the reviewer’s concerns and/or if there is anything else we can add to make them even more enthusiastic about the manuscript. Thanks again!

---

> > ### Comment · Reviewer_UMX4 · 2023-08-16
> > **Highlight changes**
> >
> > Thank you for the response and the updated manuscript. It would be very helpful if the updated manuscript highlighted the parts that were modified from the initial submission.

---

> > > ### Author Response · Authors · 2023-08-16
> > >
> > > Certainly. I've uploaded a version with major additions highlighted. There were a number of sentence-level edits + additional citations intended to reduce the manuscript's previous emphasis on protein structure prediction that I didn't highlight to avoid clutter; please let me know if you'd like to see those included as well.

---

> > > > ### Comment · Reviewer_UMX4 · 2023-08-24
> > > >
> > > > Thank you for the updates. My concern about applications to the broad ML community are mostly resolved.
> > > >
> > > > There remains this concern:
> > > >
> > > > > Apart from those in the ML community working on AF-style folding and bioinformaticians it is not directly clear how the broader ML community can easily make use of this data.
> > > >
> > > > On this note, though it will now be clear to readers with the updated manuscript that MSAs are useful in many tasks, I encourage the authors to briefly discuss the technical means by which someone fairly new to bioinformatics but with ML experience could make use of this data. e.g. point to methods for representing MSAs, or tools for loading them into deep learning frameworks, etc. As it stands, the dataset is simply shared in the format given by the alignment tool so I worry there still remains a technical hurdle to overcome for wide adoption.
> > > >
> > > > Just a small detail. The MSA in Fig 1. could be fine tuned a bit. The second sequence from top to bottom is shifted to the right slightly. The authors could also consider including an axis to indicate the sequence position and highlight conserved positions with colors as is usually done when presenting MSAs.

---

> > > > > ### Author Response · Authors · 2023-08-26
> > > > >
> > > > > I've updated the manuscript. I've replaced my homemade LaTeX MSA visualization in Figure 1 with a proper one (aligned, w/ highlights and sequence position axis). I've also added a footnote on page 6 explaining the A3M format. Note that A3M is a plaintext format and not particular to any alignment tool; with the information in the footnote, it should now be fairly straightforward even for non-bioinformatic practitioners to load the MSAs into their pipelines.
> > > > >
> > > > > We thank the reviewer again for their constructive suggestions and hope we have addressed their concerns.

---

> > > > > > ### Comment · Reviewer_UMX4 · 2023-08-26
> > > > > >
> > > > > > Looks great. I have updated my score to a 7.

---

### Decision · Program_Chairs · 2023-09-22

**Decision:**

Accept (Poster)

**Comment:**

This paper presents OpenProteinSet a database of pre-computed MSAs (aligned proteins) that is critical for numerous protein ML tasks.
The paper does a great job at explaining the value of pre-computed MSA databases and introduces the largest public MSA protein dataset, which was already successfully used to train OpenFold. The authors also did a great job at addressing the reviewer concerns.